



# 1 Insights into the role of dicarboxylic acid on CCN activity:
# 2 implications for surface tension and phase state effects

Chun Xiong[1], Binyu Kuang[1], Xiaolei Ding[3], Xiangyu Pei[1], Zhengning Xu[1], Huan Hu[3*], Zhibin Wang[1,2,4*]
[1]College of Environmental and Resource Sciences, Zhejiang University, Zhejiang Provincial Key Laboratory of Organic
Pollution Process and Control, Hangzhou, China
[2]ZJU-Hangzhou Global Scientific and Technological Innovation Center, Hangzhou, China
[3]Zhejiang University-University of Illinois at Urbana-Champaign Institute, International Campus, Zhejiang University,
Haining 314400, China
[4]Key Laboratory of Environment Remediation and Ecological Health, Ministry of Education, Zhejiang University, Hangzhou,
China
*Correspondence to*: Zhibin Wang (wangzhibin@zju.edu.cn) and Huan Hu (huanhu@intl.zju.edu.cn)
**Abstract.** Dicarboxylic acids are ubiquitous in atmospheric aerosol particles, but their roles as surfactants in cloud
condensation nuclei (CCN) activity remain unclear. In this study, we investigated CCN activity of inorganic salt (sodium
chloride and ammonium sulfate) and dicarboxylic acid (including malonic acid (MA), phenylmalonic acid (PhMA), succinic
acid (SA), phenylsuccinic acid (PhSA), adipic acid (AA), pimelic acid (PA) and octanedioic acid (OA)) mixed particles with
varied organic volume fraction (OVF), and then directly determined their surface tension and phase state at high relative
humidity (over 99.5%) by atomic force microscopy (AFM). Our results showed that CCN derived $\kappa_{CCN}$ of studied dicarboxylic
acids ranged in 0.003-0.240. A linearly positive relation between $\kappa_{CCN}$ and solubility was obtained for slightly dissolved species,
while negative relation was found between $\kappa_{CCN}$ and molecular volume for highly soluble species. For most inorganic
salt/dicarboxylic acid (MA, PhMA, SA, PhSA and PA), a good closure within 30% relative bias between $\kappa_{CCN}$ and chemistry
derived $\kappa_{Chem}$ were obtained. However, $\kappa_{CCN}$ values of inorganic salt/AA and inorganic salt/OA systems were surprisingly 0.3-
3.0 times higher than $\kappa_{Chem}$, which was attributed to surface tension reduction as AFM results showed that their surface tensions
were 20%-42% lower than that of water (72 mN m$^{-1}$). Meanwhile, semisolid phase states were obtained for inorganic salt/AA
and inorganic salt/OA and may also affect hygroscopicity closure results. Our study highlights that surface tension reduction
should be considered to investigate aerosol-cloud interactions.

## 26 1 Introduction

Atmospheric particles can indirectly affect global climate through their impact on aerosol-cloud interaction by serving as cloud
condensation nuclei (CCN) (Rosenfeld et al., 2014). Exploring the factors affecting CCN activation could help to understand
the aerosol-cloud interactions and thus decrease the uncertainty in the assessment of climate model. Köhler theory provides
the basis for linking CCN activity with aerosol thermodynamic properties (Köhler, 1936), in which size and chemical
composition are key factors to determine the activation of aerosol particles. Previous studies pointed out that aerosol number



size distribution is essential to determine CCN concentration other than composition (Dusek et al., 2006; Gunthe et al., 2009;
Rose et al., 2010). The role of particle chemistry in the activation process, however, is still debatable due to the complexity of
chemical constitution.
Single parameter $\kappa$ was introduced in Köhler theory to describe hygroscopicity of aerosol particles (Petters and Kreidenweis,
2007). $\kappa$-Köhler theory usually performed well in predictions of hygroscopicity and CCN number concentration (Rose et al.,
2010; Kawana et al., 2016; Cai et al., 2020; Zhang et al., 2020). However, remarkable offset was also found because of the
simplifications in $\kappa$-Köhler theory (Ruehl et al., 2016; Ovadnevaite et al., 2017). For example, aerosol droplet is assumed to
be diluted near activation and surface tension is usually simply treated as that of pure water, which is sometimes not reasonable
in the presence of atmospheric surfactants (Lowe et al., 2019). Yet many previous studies investigated surface tension effect
of atmospheric surfactant on aerosol CCN activity (Ruehl and Wilson, 2014; Ruehl et al., 2016; Ovadnevaite et al., 2017). At
Mace Head, Ovadnevaite et al. (2017) observed significant underestimation of CCN number concentration (one tenth) in a
nascent ultrafine mode event with high organic mass fraction (55%). The underestimation was improved by applying lower
water surface tension (~ 68% of water surface tension). For surfactant sodium octyl sulfate, Peng et al. (2022) found that CCN-
derived $\kappa_{CCN}$ was around 2.4 times larger than growth factor derived $\kappa_{GF}$, which was ascribed to surface tension reduction and
solubility limit. Though established thermodynamic models considering surface tension reductions such as compressed film
model (Ruehl et al., 2016) and liquid-liquid phase separation model (Ovadnevaite et al., 2017; Liu et al., 2018) explained the
discrepancies of CCN activity or CCN number concentration closure, dataset of direct measurement of surface tension for
submicron particles are very rare.
Dicarboxylic acids are ubiquitous in atmospheric aerosol particle as a main contributor to organic aerosol mass (mass
contribution to total particulate carbon could exceed 10% in remote area) (Römpp et al., 2006; Ho et al., 2010; Hyder et al.,
2012). Primary emission (e.g. biomass burning and fossil fuel combustion) and secondary formation (e.g. photooxidation of
unsaturated fatty acids) were major sources of dicarboxylic acids (Ho et al., 2010). Furthermore, dicarboxylic acids are also
known as important atmospheric surfactants and their surface activities in water solutions showed a positive relation with
carbon number (Aumann et al., 2010). Currently, most studies investigated surface tension effect of dicarboxylic acids on CCN
activation by measuring surface tension of their solutions and using models based on solution results (Lee and Hildemann,
2013, 2014; Ruehl et al., 2016; Zhang et al., 2021; Vepsäläinen et al., 2022). However, the values derived from bulk solutions
may not be a reasonable represent for aerosol particles because their high surface-to-volume ratio may affect the distribution
of surfactant between surface and bulk (Ruehl et al., 2010; Ruehl and Wilson, 2014). Recently, new methods of surface tension
measurement for particles were introduced such as microfluid (Metcalf et al., 2016) and optical tweezers (Bzdek et al., 2020),
but their samples were micrometre size droplets. Morris et al. (2015) presented a way to directly measure surface tension of
submicron particles under controlled relative humidity (RH) by atomic force microscopy (AFM). Later, AFM was further
reported to be an important tool to probe phase state of individual particles (Lee et al., 2017a; Lee et al., 2017b; Lee and
Tivanski, 2021). However, most measurements using AFM were performed with RH under 95% (Morris et al., 2015; Lee et



al., 2017b; Ray et al., 2019; Lee et al., 2020) but rare in higher RH conditions. When RH approaches 100%, Kelvin effect
becomes comparable to the Raoult effect in controlling hygroscopicity, so measurements around 100% RH can help resolve
discrepancies between sub-saturated hygroscopicity and CCN activity (Ruehl and Wilson, 2014).
In this study, we firstly measured CCN activities of internal mixtures containing inorganic salt and dicarboxylic acid. Then,
we directly obtained their surface tension and phase states by AFM under relatively high RH (over 99.5%). Our results could
provide directly dataset of surface tension and phase state of inorganic salts-dicarboxylic acids internal mixed particles, which
would help to decrease the uncertainty for climate models.
**2 Methods**
**2.1 Experiments**
**2.1.1 Chemicals**
Nine used compounds in the present study were sodium chloride (NaCl), ammonium sulfate (AS), malonic acid (MA),
phenylmalonic acid (PhMA), succinic acid (SA), phenylsuccinic acid (PhSA), adipic acid (AA), pimelic acid (PA) and
Octanedioic acid (OA). Their relevant properties investigated in this study were summarized in **Table 1**.
**2.1.2 CCN activity measurements**
The measurement setup is shown in **Fig. 1**. In brief, particles containing single and mixed chemicals were generated with water
solutions (~ 1‰) by a constant output atomizer (TSI 3079A). After drying (RH < 15%), monodispersed aerosol particles were
obtained by differential mobility analyzer (DMA, TSI 3081) with the sheath to sample flow ratio of 10, and then were split
between a condensation particle counter (CPC, TSI 3772) for measuring number concentration of total particles ($N_{CN}$) and a
Cloud Condensation Nuclei Counter (CCNC, DMT-200) for measuring number concentration of CCN ($N_{CCN}$).
In this study, the CCNC was operated in Scanning Flow CCN Analysis (SFCA) mode, which was introduced elsewhere (Moore
and Nenes, 2009). In short, the pressure and $\Delta T$ of CCNC were kept constant, the flow rate was continuously and linearly
varied from 0.2 L min$^{-1}$ to 1 L min$^{-1}$ or vice versa (1-0.2 L min$^{-1}$) within 125 s and the interval time for stabilization is 25 s.
The supersaturations in CCNC was calibrated under four $\Delta T$ (4K, 6K, 10K and 18K). We obtained sigmoidal curves of
activation ratio ($N_{CCN}/N_{CN}$) versus flow rate, then fitted the inflection point of the curves as critical flow rate $Q_{50}$. Ammonium
sulfate was used to determine supersaturation ratio with an activity parameterization Köhler model AP3 as suggested by Rose
et al. (2008). The calibration results were showed in **Fig. S1**.
**2.1.3 Surface tension measurements**
As showed in **Fig.1**, samples for AFM analysis were collected through deposition by impaction with an eight stage non-viable
particle sizing sampler (Models BGI20800 Series, BGI Incorporation) onto hydrophobically silicon wafers (Ding et al., 2020).



The aerodynamic size of collected particles was ranged in 0.4 μm-1 μm (50% efficiency). The substrate deposited particles
were stored under dry condition (RH < 10%) and most of the samples were studied at the same day to avoid possible sample
aging.
Surface tension measurement was performed using an AFM system (Cypher ES, Asylum Research). Cypher ES contains a
small cell with air inlet and outlet, it enables to scan samples under different environmental conditions such as RH. RH in cell
was achieved and maintained by humidified flow. RH in cell was measured by a RH sensor (SHT 85, Sensirion Inc.). Custom-
built high aspect ratio (HAR) platinum AFM probes with constant diameter and nominal spring constant of ~ 3.0 N m$^{-1}$ were
used for particle imaging and surface tension measurements (**Fig. S2**) (Morris et al., 2015). The platinum nanoneedles could
well measure surface tension of pure water and 1, 3-propanediol (**Fig. S3**). The procedures of making nanotips were detailly
described in a manuscript under review and a brief description was given here. Firstly, dual-beam-focused ion beam (FIB,
ZEISS crossbeam 350) microscope was used to etch the top of the tip (Multi75Al-G purchased from BudgetSensors Inc.),
making the etched tip flat. Then, FIB was used to deposit a cylindrical metal platinum column (100 nm-500 nm diameter) on
the flat surface of the etched tip.
The principles of surface tension measurement using AFM were described elsewhere (Yazdanpanah et al., 2008; Morris et al.,
2015; Lee et al., 2017a). Collected samples were firstly imaged in tapping mode to locate individual particles under dry
condition (RH < 10%), then the RH gradually increased to over 99.5% in ~ 40 minutes (**Fig.S4**). Force-distance plots of droplet
were obtained by contact mode. A tip velocity of 1-2 μm s$^{-1}$ and dwell time of 1-2 seconds were used for all measurements
(Kaluarachchi et al., 2021). More than 10 force plots were collected on at least 5 individual droplets. Precise diameter of
nanoneedle was calibrated by measuring surface tension of pure water by adding a water droplet (2-3 mm height) onto silicon
wafer (Kaluarachchi et al., 2021). New probe was used for different chemicals in order to avoid possible contamination of the
AFM probe.
**2.2 Theory**
Based on $\kappa$-Köhler theory, hygroscopicity parameter $\kappa_{CCN}$ can be calculated by:
$$\kappa_{CCN} = \frac{4A^3}{27D_d^3 ln^2(1+s_c)}, A = \frac{4M_w\sigma_w}{RT\rho_w}$$    (1)
where $\sigma_w$, $M_w$ and $\rho_w$ are surface tension, molecular weight and density of water, respectively. $R$ is universal gas constant and
$T$ is temperature (298.15K). $s_c$ is critical supersaturation ratio. $D_d$ is dry diameter. In addition, hygroscopicity $\kappa$ of
multicomponent chemical system can also be calculated assuming a Zdanovskii, Stokes, and Robinson (ZSR) simple mixing
rule. $\kappa$ based on the chemical composition ($\kappa_{Chem}$) of mixed aerosol was calculated by:
$$\kappa_{Chem} = OVF \cdot \kappa_{org} + (1 - OVF) \cdot \kappa_{inorg} ,$$    (2)



where $\kappa_{org}$ and $\kappa_{inorg}$ are hygroscopicity $\kappa$ values (here obtained $\kappa_{CCN}$ values were used) of single organic acids and inorganic
salts.
As described by Morris et al. (2015), the basis of surface tension measurement for a liquid droplet by AFM was calculated by:
$\sigma = \frac{F_r}{2\pi r}$,         (3)
where $F_r$ is the retention force to break the meniscus by the tip of AFM probe, $r$ is the radius of the AFM probe tip, and $\sigma$ is
surface tension of the droplet. The retention force is the force difference before and after the probe was just retracted from the
droplet.

## 3 Results and discussion

### 3.1 $\kappa_{CCN}$ of single component

$\kappa_{CCN}$ values for single component aerosols were summarized in **Table 2**. $\kappa_{CCN}$ of NaCl, AS, MA, SA and AA were 1.325 ±
0.038, 0.562 ± 0.059, 0.240 ± 0.036, 0.204 ± 0.023 and 0.008 ± 0.001, respectively, being consistent with previous results
(Petters and Kreidenweis, 2007; Kuwata et al., 2013). $\kappa_{CCN}$ of PA and OA were 0.112 ± 0.010 and 0.003 ± 0.0002, which were
20% lower and twice higher than those reported by Kuwata et al. (2013). Possible factor may be the purity of solutes, because
additional hydrophobic (or hygroscopic) matters in commercial reagents may possibly decrease (increase) organic
hygroscopicity (Hings et al., 2008). $\kappa_{CCN}$ values of PhMA and PhSA were 0.183 ± 0.032 and 0.145 ± 0.017, respectively,
which to our knowledge are firstly reported in this study.
Solubility and molar volume of dicarboxylic acids were essential factors influencing their hygroscopicity (Kumar et al., 2003;
Han et al., 2022). In this study, we considered two regimes: highly soluble organic components (with water solubility over 100
g L$^{-1}$) and slightly soluble organic components (with water solubility between 10-100 g L$^{-1}$), which was consistent with
previous study (Kuwata et al., 2013). As showed in **Fig. 2a**, the $\kappa_{CCN}$ values for highly soluble components decreased linearly
with increased molecular volumes. This trend was similar to $\kappa_{CCN}$ values for sugar as well as dicarboxylic acids reported by
Chan et al. (2008). In **Fig. 2b**, $\kappa_{CCN}$ values of sparely soluble components (AA, PA, SA and OA) showed an increased trend
with solubility, as organic matter with the higher water solubility would dissolve more and have a higher molar concentration,
resulting in reduction in water activity and higher hygroscopicity (Luo et al., 2020; Han et al., 2022).
Organic functional group could also affect hygroscopicity (Suda et al., 2014; Petters et al., 2017). $\kappa_{CCN}$ of PA (0.112) was
higher than those of AA (0.008) and OA (0.003), which is contrary to results in Suda et al. (2014) and Petters et al. (2017) that
hygroscopicity decreased with increased number of methylene. This phenomenon was attributed to the odd-even effect of
dicarboxylic acids, that is, diacids with odd numbers of carbon atoms being more soluble than those with adjacent even
numbers (Zhang et al., 2013). Furthermore, $\kappa_{CCN}$ values of PhMA and PhSA were both lower than that of MA and SA,
respectively, indicating that the addition of phenyl showed negative effectes on hygroscopicity. The addition of phenyl



substitution increased the molar volumes of MA and SA and may contribute to the drops of hygroscopicity (Petters et al.,
154  2009).

### 3.2 $\kappa_{CCN}$ of inorganic salt-dicarboxylic acid mixed components

**Figure 3** presents the $\kappa_{CCN}$ values of inorganic salt/dicarboxylic acid mixed particles with varied organic volume fractions
(OVF). Overall, $\kappa_{CCN}$ of each inorganic salt/dicarboxylic acid system showed a decreased trend with increased OVF. For
example, $\kappa_{CCN}$ of AS/MA particles with OVF of 57%, 73% and 88% were 0.399, 0.373 and 0.336, respectively. Larger fractions
of dicarboxylic acids (with low hygroscopicity compares to inorganic salts) caused more decrease in hygroscopicity of
inorganic/dicarboxylic acid system. As for inorganic salt/dicarboxylic acid systems with same OVF,     $\kappa_{CCN}$ values of systems
of AS/MA, AS/SA, AS/PhMA, AS/PhSA and AS/PA with 57% OVF were 0.399, 0.382, 0.364, 0.340 and 0.334, following
the order of $\kappa_{CCN}$ values of single dicarboxylic acid (**Fig. 3a**). However, $\kappa_{CCN}$ values of NaCl/AA and NaCl/OA mixed particles
with OVF of 60% were 0.734 and 0.685, even higher than that of NaCl/MA (0.639), demonstrating an opposite trend with
respect to those of single components. This discrepancy could be ascribed to surface tension reduction because AA and OA
showed different physical properties (e.g. deliquescence point, surface activity and solubility) when comparing with the other
organics, thus may result in distinct microphysics processes during interactions with inorganic salts and water content. AA and
OA own lowest solubilities and high deliquescence RH (**Table1**) among experimental dicarboxylic acids, which potentially
lead to their weak CCN activities (Hings et al., 2008). However, inorganic salts were found to facilitate the deliquescence of
dicarboxylic acid (Bilde and Svenningsson, 2004; Sjogren et al., 2007; Minambres et al., 2013). AS/AA mixed particles
deliquescence under 78%-83% RH with mass fractions of AA between 50%-80% (Sjogren et al., 2007). Small amount of NaCl
(2% mass faction) could notably decrease $s_c$ of AA with 80 nm dry diameter from over 2% to ~0.6% (Bilde and Svenningsson,
2004). Thus, addition of inorganic salts facilitates deliquescence of OA and AA under lower RH, which may further promote
phase state transition from solid to liquid (or semisolid) and cause surface tension reductions as OA and AA show stronger
surface activities than most of the rest dicarboxylic acids because of longer carbon chain (Aumann et al., 2010). This indication
was further confirmed by AFM surface tension measurement, as discussed in Section 3.4.

### 3.3 Closure study between $\kappa_{CCN}$ and $\kappa_{Chem}$

$\kappa_{CCN}$ and $\kappa_{Chem}$ values for inorganic salt/dicarboxylic acid mixed particles were showed in **Fig. 4**. $\kappa_{CCN}$ values of inorganic salt
and most dicarboxylic acids (MA, PhMA, SA, PhSA and PA) mixed particles could be predicted by ZSR mixing rule with
relative difference below 30% (**Fig. 4a**). Similar results have been found in previous lab and filed studies (Ruehl et al., 2012;
Kuwata et al., 2013; Wu et al., 2013; Dawson et al., 2016; Nguyen et al., 2017; Ovadnevaite et al., 2017), indicating that semi-
experimental ZSR mixing rule could be a useful method to predicted mixed particles hygroscopicity and CCN activation. For
instance, Dawson et al. (2016) reported consistence between $\kappa_{CCN}$ and $\kappa_{Chem}$ for NaCl/xanthan gum and CaCO$_3$/xanthan gum
mixed particles within 10% uncertainty. Wu et al. (2013) also obtained same closure results in a field study at central Germany,



for particles containing 60%-80% organic mass fraction and 30%-50% inorganic salts. Meanwhile, CCN studies also found
that using $\kappa_{Chem}$ could well predict measured CCN number concentration (Juranyi et al., 2010; Rose et al., 2010; Almeida et
al., 2014; Kawana et al., 2016; Cai et al., 2020; Zhang et al., 2020). However, for inorganic/AA and inorganic/OA mixed
particles (**Fig. 4b)**, their $\kappa_{CCN}$ values were 0.3-3.0 times higher than $\kappa_{Chem}$. Surface tension reduction was one of the potential
causes, as discussed in section 3.2 that OA and AA with strong surface activity and low solubilities may result in stronger
surface tension reduction than most of the rest dicarboxylic acids. In addition, the underprediction showed a gradual increased
trend with increased OVF since increased OVF lead to higher concentration of organics, thus leading to more surface tension
reduction. Surface tension reduction in water solution caused by atmospheric surfactants were observed frequently in previous
studies (Facchini et al., 1999; Gerard et al., 2016). Results have showed that neglect of surface tension reduction may lead to
higher $\kappa_{CCN}$ values than $\kappa_{Chem}$ or growth factor derived $\kappa_{GF}$ (Irwin et al., 2010; Wu et al., 2013; Zhao et al., 2016; Hu et al.,
2020; Peng et al., 2021), as well as underpredictions of CCN number concentration (Good et al., 2010; Asa-Awuku et al., 2011;
Ovadnevaite et al., 2017; Cai et al., 2020). Hu et al. (2020) reported that $\kappa_{Chem}$ underpredicted $\kappa_{CCN}$ by 13% and 18% at
supersaturation ratios of 0.1% and 0.3%, which may be attributed to the depression of droplet surface tension by potential
surface-active organics. Likewise, Ovadnevaite et al. (2017) only predicted one tenth of measured CCN number concentration
in a nascent ultrafine mode event because of the surface tension reduction, and the notable underestimation was improved by
applying lower water surface tension (~ 68% of water surface tension) in $\kappa$-Köhler theory.
Apart from surface tension reduction, aerosol phase states could also bring uncertainty to critical supersaturation and
hygroscopicity predictions (Henning et al., 2005; Hodas et al., 2015; Peng et al., 2016; Zhao et al., 2016). Being different from
tradition Köhler curve with only one maximum, modified Köhler curve for inorganic salt and slightly soluble dicarboxylic
acid (e.g. AA) mixed particles accounting for limited solubility obtained two maxima of critical supersaturation ratios and the
higher value among the two maxima determined CCN activation (Bilde and Svenningsson, 2004). The maximum at the larger
wet diameter is identical with that obtained by assuming that the organic acids are infinitely soluble in water (i.e. classical
Köhler theory). And the other maximum with smaller wet diameter represents the point that all slightly soluble material is
fully dissolved and the maximum can also be viewed as an activation barrier which is due to the presence of a undissolved
solid part of organic acid (Henning et al., 2005). Pajunoja et al. (2015) reported that biogenic secondary organic aerosol (SOA)
particles formed from isoprene showed an increased trend of hygroscopicity parameter from 0.05 to nearly 0.15 when RH
increased from 40% to supersaturation. They indirectly found the biogenic SOA to be semisolid phase thus the increased trend
of hygroscopicity $\kappa$ was explained by the gradual phase transition from solid to semisolid (or liquid) with raised RH because
water content may gradually wet and dissolve the organic surface and form water film (Pajunoja et al., 2015). The phase
transition (or water film formation) of pure OA and AA would be difficult (i.e. high RH is required) because of their high
deliquescence point and low solubilities, but could be easier (i.e. required high RH is decreased) by addition of inorganic salts.
Overall, phase state and surface tension of atmospheric aerosol were two essential factors influencing their hygroscopicity and



CCN activation. Though there are several indirect ways detecting aerosol phase state (Pajunoja et al., 2015; Shiraiwa et al.,
2017), current studies about directly measurements are still very limited.

**3.4 Phase state and surface tension of inorganic salt/dicarboxylic acid mixed particles**

**3.4.1 Phase state**

We obtained phase states of inorganic salt/dicarboxylic acid under high RH environment (over 99.5%) by analyzing shapes of
force plot based on AFM system (Lee et al., 2017a; Lee and Tivanski, 2021). **Figure 5a** showed force plot of NaCl/MA mixed
particles with 75% OVF. AFM probe needle tip approached the droplet vertically before contacting with droplet, needle tip
was not disturbed by extra force (red line). Then, needle tip came in contact with the droplet, resulting in an abrupt negative
force (i.e. needle was attracting by drop). After that, needle moved through the droplet with negative force until contacting
with the substrate. When tip contacted substrate, the negative force would quickly be positive (repulsive force), exceeding a
predefined maximum amount of force. Then the tip retracted back away from the droplet, as indicates by blue line. Because
of the surface tension of droplet surface, needle tip would experience attractive force and abruptly turned to zero when tip
separated from droplet surface. Our observation in **Fig. 5a** showed a similar shape with results reported by Morris et al. (2015),
indicating the particles were liquid. Most of the studied inorganic salt/dicarboxylic acid (MA, PhMA, SA, PhSA and PA) were
liquid under RH over 99.5%.
However, for AS/SA (72% and 88% OVF), NaCl/AA (89% OVF), AS/AA (57%, 72% and 88% OVF) and AS/OA (88%
OVF), the shape force plots were totally different. During the tip contacting with particle, force plots showing a jagging profile,
as shown in **Fig. 5b.** This shape is nearly the same as the curves for NaBr particles under 52% RH reported by Lee et al.
(2017a). They explained the phase of NaBr was semi-solid and jagging profile in tip approaching was caused by its viscosity.
Therefore, AS/SA (72% and 88% OVF), NaCl/AA (89% OVF), AS/AA (57%, 72% and 88% OVF) and AS/OA (88% OVF)
mixed particles were indicated to be semisolid. Semisolid phase states were more likely to occur when containing higher OVF
of dicarboxylic acids with lower solubilities and higher deliquescence point (SA, AA and OA) and inorganic salts with
comparative lower hygroscopicity (AS), as in this circumstance water content may be insufficient and could not easily dissolve
organics. Therefore, semisolid phase of inorganic salt/AA and inorganic salt/OA mixed particles provides evident for phase
state effect on aerosol hygroscopicity, which may attribute to higher $\kappa_{CCN}$ than $\kappa_{Chem}$ as discussed in section 3.3 (**Fig 4b**).
Though AS/SA mixed particles (72% and 88% OVF) were semisolid because of high deliquescence point (98%) of SA, their
good closure between $\kappa_{CCN}$ and $\kappa_{Chem}$ may ascribe to higher solubility of SA, which may intensify the water absorption after
deliquescence thus phase transition from semisolid to diluted liquid when activating to CCN.

**3.4.2 Surface tension**

Lee et al. (2017a) pointed out that surface tension calculation could not be achieved for semisolid particles, because the
measured retention force was not solely attributed to surface tension, but have additional contributions that include viscosity.





Therefore, only surface tensions of inorganic salt/dicarboxylic acid mixed particles that were liquid were further obtained by
**Eq.3**. Surface tension results were summarized in **Fig. 6**. Overall, surface tensions of all inorganic salt/dicarboxylic acid mixed
particles showed a decrease trend with increased OVF as higher OVF may result in higher organic solute concentrations thus
caused more surface tension reduction. Surface tensions of inorganic salts mixed with MA, PhMA, SA, PhSA and PA lowered
by within 12% than that of pure water (72 mN m$^{-1}$), indicating that droplets got strongly diluted at RH over 99.5%, and ought
to be more diluted when activation occurs. This may contribute to $\kappa$ closure within 30% deviation in **Fig. 4a** because diluted
solution and water surface tension were assumed in $\kappa$-Köhler theory. However, surface tensions of inorganic salts/AA and
inorganic salts/OA mixed particles showed notable reductions (20%-42%), which may contribute to their higher $\kappa_{CCN}$ values
than $\kappa_{Chem}$ **(Fig. 4b)**. Besides, notable surface tension reductions of particles containing OA or AA indicated that organic
solubility plays an important role in surface tension reduction as AA and OA have the lowest solubilities among studied
dicarboxylic acids. Besides, OA and AA own higher deliquescence point and longer carbon chains than most of the rest studied
organics and thus deliquescence RH and strong surface activity are also essential factors attributing to surface tension reduction
for inorganic salt/dicarboxylic acid mixed particles. Furthermore, for dicarboxylic acids, lower organic solubilities may be
more important factor causing surface tension reduction than deliquescence RH and surface activity. This was because PA
with higher solubility, but similar deliquescence RH and surface activity like AA and OA did not show much depression of
surface tension when mixed with inorganic salts.

## 263     4 Conclusions

The role of surfactants such as dicarboxylic acids in CCN activity were often ignored in aerosol hygroscopicity studies and
currently climate models. In this study, we analyzed CCN activities of inorganic salt/dicarboxylic acid internal mixed particles
with varied OVF and directly measured their phase state and surface tension by AFM under relative high RH.
$\kappa_{CCN}$ values of single dicarboxylic acid located in the range of 0.003-0.240. A linearly positive relation between $\kappa_{CCN}$ and
solubility were obtained for slightly dissolved species, while negative relation was found between $\kappa_{CCN}$ and molecular volume
for highly soluble species. $\kappa_{CCN}$ of PhMA and PhSA were lower than those of MA and SA, respectively, revealing that addition
of phenyl radical could weaken hygroscopicity of dicarboxylic acid.
For most inorganic salt/dicarboxylic acid (MA, PhMA, SA, PhSA and PA), $\kappa_{CCN}$ of mixed particles with same OVF showed
an overall decrease trend and followed the order of $\kappa_{CCN}$ values of single dicarboxylic acid. Good closure within 30% relative
bias between $\kappa_{CCN}$ and $\kappa_{Chem}$ were obtained. On the contrast, our results demonstrated that the semisolid phase state and surface
tension reduction (20%-42%) are the potential factors to explain the enhanced CCN activity of inorganic salts/OA and
inorganic salts/AA mixed particles. Slightly dissolved dicarboxylic acids with lower solubilities, higher deliquescence point
and surface activity are more likely to cause notable surface tension depression for inorganic salt/dicarboxylic acid mix
particles. Therefore, we proposed that surface tension reduction and phase state should be carefully considered in future models



and observations, especially for slightly soluble organics with lower solubilities, high deliquescence RH and strong surface
activity.

***Data availability.*** The data used in this paper can be obtained from the corresponding author upon request.
***Author contributions.*** CX did the experiments, analyzed data, plotted the figures and wrote the original draft. BYK contributed
data analyzing and discussion, reviewed the manuscript and contributed to fund acquisition. XLD and XYP contributed to the
instrumentation and discussion. ZNX contributed to the discussion and fund acquisition. HH contributed to the instrumentation,
discussion and fund acquisition. ZBW administrated the project, conceptualized the study, reviewed the manuscript and
contributed to fund acquisition.
***Acknowledgment.*** The research was supported by National Natural Science Foundation of China (91844301, 42005087,
61974128 and 42005086) and the Fundamental Research Funds for the Central Universities (2018QNA6008). We appreciate
Shikuan Yang, Qianqian Ding and Xueyan Chen for making and kindly sharing hydrophobically silicon wafers. We likewise
thank Ren Zhu, Lin Liu, Renwei Mao and Yuzhong Zhang for the discussions about AFM experiment.
***Competing interests.*** The authors declare no competing financial interest.

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



**Table 1. Substances and their relevant properties investigated in this study.**

| Compounds | Molar weight (g mol$^{-1}$) | Density (g cm$^{-3}$) | Solubility (g L$^{-1}$) | DRH (%RH) | Purity | Supplier |
|---|---|---|---|---|---|---|
| NaCl | 58.44[a] | 2.16[a] | 360[b] | 73-77[c] | GR | Sinopharm Chemical Reagent |
| AS | 132.13[a] | 1.77[a] | 770[b] | 78-82[c] | ≥99% | Sigma Aldrich |
| MA | 104.06[a] | 1.63[a] | 1400[b] | 65-76[c] | ≥99% | Sigma Aldrich |
| PhMA | 180.16[a] | 1.40[a] | 131[a] | NA | 98% | Aladdin |
| SA | 118.09[a] | 1.57[a] | 80[b] | 98[d] | ≥99% | Aladdin |
| PhSA | 194.19[a] | 1.13[a] | 241[a] | NA | 98% | Macklin |
| AA | 146.14[a] | 1.36[a] | 14.4[b] | ~100[e] | GR | Sinopharm Chemical Reagent |
| PA | 160.17[a] | 1.28[a] | 25[b] | >90[c] | 99% | Macklin |
| OA | 174.20[a] | 1.16[a] | 12[a] | >90[c] | 99% | Aladdin |

[a] https://comptox.epa.gov/ (last access: 3rd August 2022). [b] https://www.chemicalbook.com/ (last access: 3rd August 2022). [c]
Peng et al. (2022) and references therein. [d] Peng et al. (2001). [e] Parsons et al. (2004). DRH means deliquescence RH. GR
means guaranteed reagent. NA indicates no reported results are available.

**Table 2. Summary of $\kappa_{CCN}$ for single component particles.**

| Chemicals | $D_d$ (nm) | $\kappa_{CCN}$ mean ± standard deviation | Previous reported $\kappa_{CCN}$ |
|---|---|---|---|
| NaCl | 50, 65, 76, 88, 100 | 1.325 ± 0.038 | 1.28[a] |
| AS | 50, 65, 76, 88, 100 | 0.562 ± 0.059 | 0.61[a] |
| MA | 50, 65, 76, 88, 100 | 0.240 ± 0.036 | 0.227[a] |
| PhMA | 50, 65, 76, 88, 100 | 0.183 ± 0.032 | This study |
| SA | 50, 65, 76, 88, 100 | 0.204 ± 0.023 | 0.166-0.295[a] |
| PhSA | 50, 65, 76, 88, 100 | 0.145 ± 0.017 | This study |
| AA | 140, 160, 180, 200 | 0.008 ± 0.001 | 0.005-0.008[b] |
| PA | 65, 76, 88, 100 | 0.112 ± 0.010 | 0.14[b] |
| OA | 200, 220, 240, 260 | 0.003 ± 0.0002 | 0.001[b] |

[a] Petters et al., 2007; [b] Kuwata et al. (2013) and references therein.






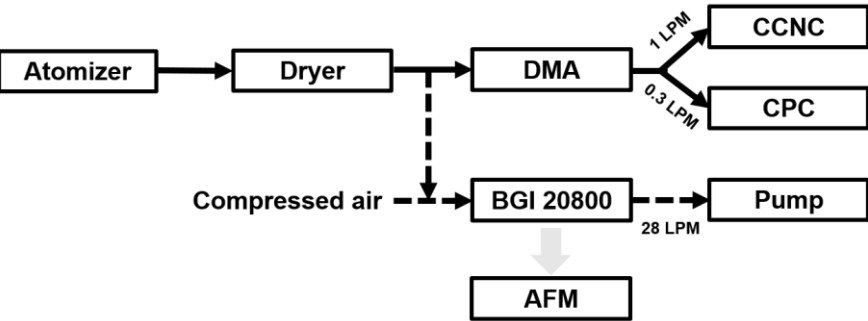


**Figure 1: Schematic illustration of the instrumental set-up. The arrow indicates the flow direction. LPM means liter per minute.**


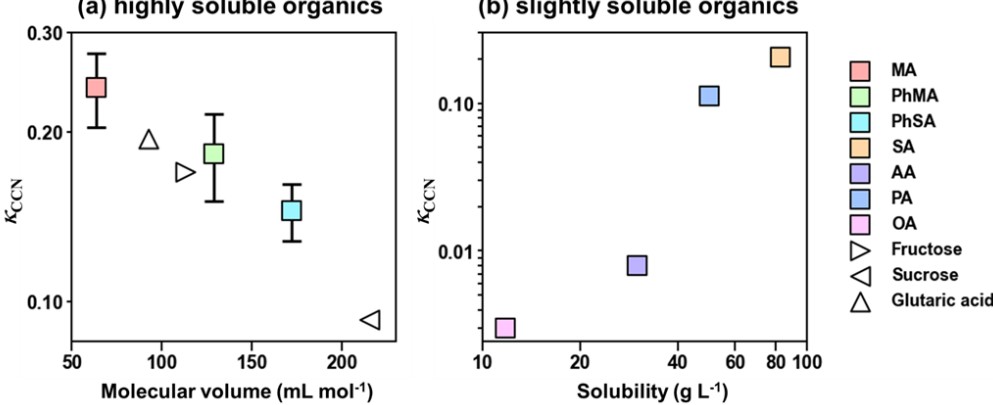


**Figure 2: $\kappa_{CCN}$ of organic compounds as a function of (a) molecular volume and (b) solubility. Solid squares represent $\kappa_{CCN}$ results in this study while hollow triangles were $\kappa_{CCN}$ results obtained from Chan et al. (2008).**


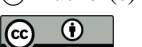



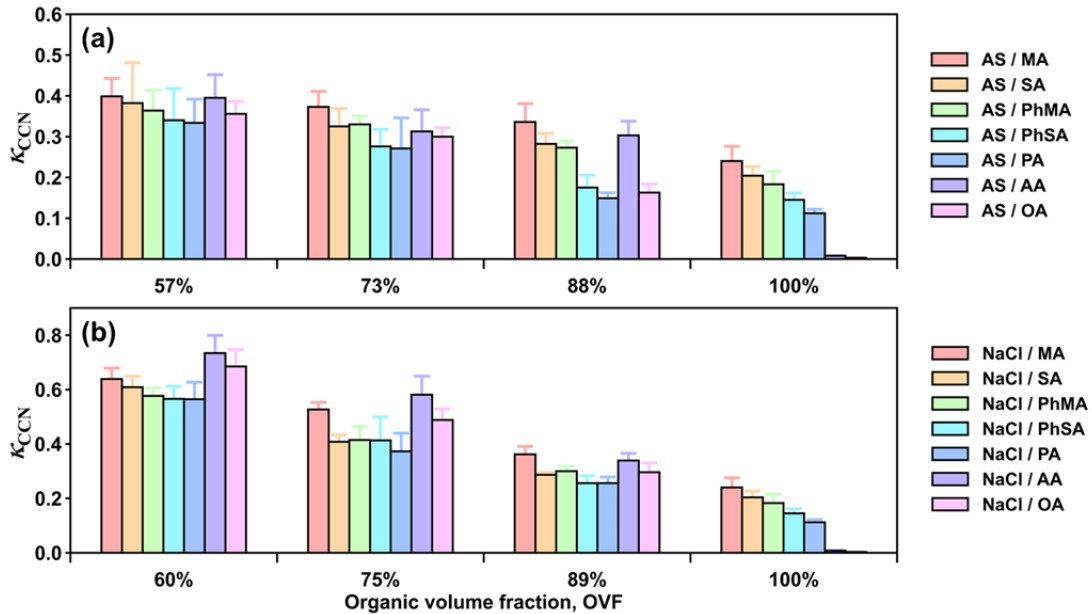

Figure 3: $\kappa_{CCN}$ of (a) AS/dicarboxylic acid and (b) NaCl/dicarboxylic acid mixed particles with varied OVF.

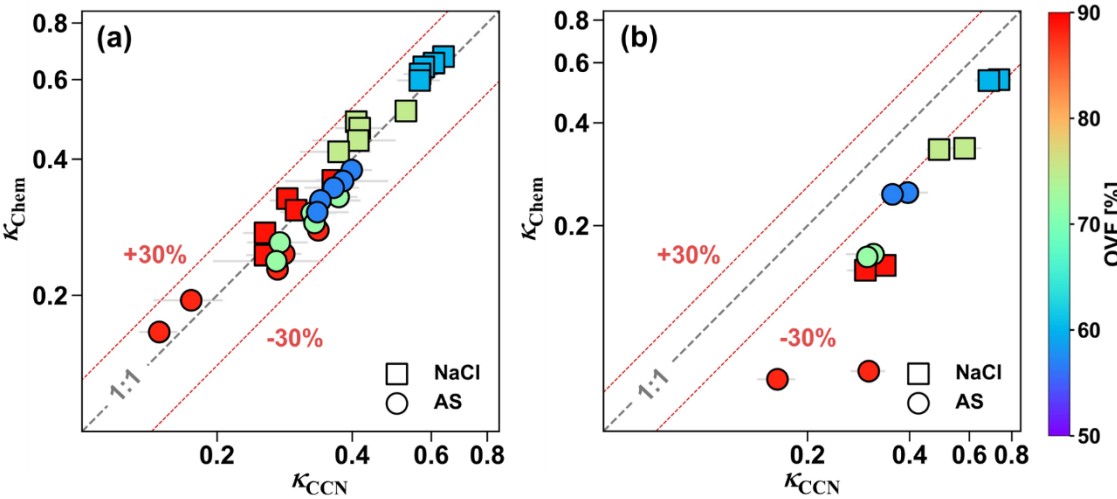

Figure 4: Comparison between $\kappa_{CCN}$ and $\kappa_{Chem}$ of (a) inorganic salt mixed with MA, PhMA, SA, PhSA and PA (b) inorganic salt mixed with AA and OA. Square represents NaCl containing particles and circle represents AS containing particles. Color bar indicates OVF.





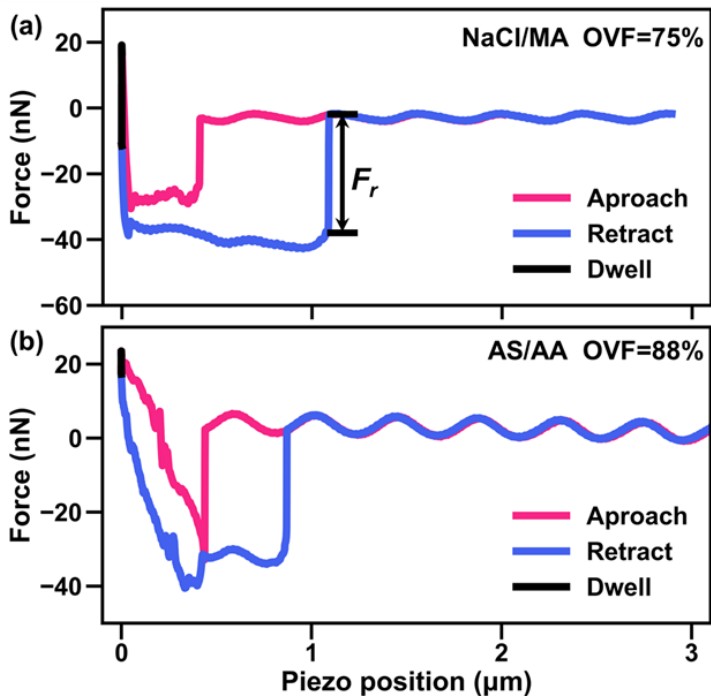

513

**Figure 5: AFM force plots of (a) NaCl/MA system with 75% OVF and (b) AS/AA system with 88% OVF. $F_r$ is the retention force to break the meniscus by the tip of AFM probe.**

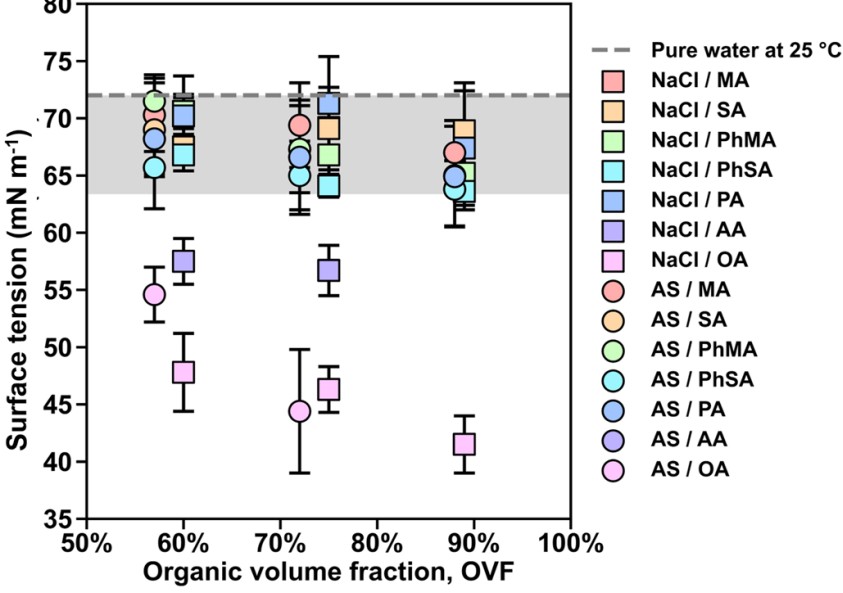

516

**Figure 6: Measured surface tension values of inorganic salt/dicarboxylic acid particles under RH over 99.5%. Gray area covers the surface tension reductions below 12% comparing with pure water (72 mN m$^{-1}$).**

519