# Peer review of "Insights into the role of dicarboxylic acid on CCN activity"

_EGUsphere, 2022_

## Author Comment (AC1)

**Reply to comments on "Insights into the role of dicarboxylic acid on CCN activity: implications for surface tension and phase state effects" by Xiong et al.**

**Reply to Anonymous Referee #1**

1) This manuscript investigated the CCN activity of the mixtures of inorganic salt and dicarboxylic acid and further determined their phase state and surface tension by using the atomic force microscopy. The results show that the bias between the $\kappa_{CCN}$ and $\kappa_{Chem}$ of inorganic salt/AA and inorganic salt/OA can be explained by the surface tension reduction. The study highlights the role of surface tension reduction in evaluating CCN activity. The study is interesting in several aspects. The topic is one that is currently undergoing debate in the atmospheric chemistry and aerosols science communities, and it is of interest to the readership of ACP. The study could be acceptable for publication if revisions are made with consideration of the issues listed below.

**Response: We truly appreciate the constructive comments and suggestions raised by the reviewer. Those comments are valuable and very helpful for improving our paper, as well as the important guiding significance to our studies. Below we provide a point-by-point response to individual comment. The responses are shown in brown and bold fonts, and the added/rewritten parts are presented in blue and bold fonts.**

**Major comments:**

2) One of my major concern is that the measured size of collected particles in this study, which ranged from 0.4-1μm. But, as has been known that, surface tension lowering effect by the surfactants was more important or obvious for fine and ultrafine particles with diameters smaller than 100 nm (Ovadnevaite et al., 2017). As particles grows, the solution is diluted and the surface tension lowering effect becomes weak. So, it would be more meaningful and or with scientific significance if the authors could look at and measure the smaller size particles.

Response: Thank you for the advice. We totally agree that the surface tension effect of surfactants on hygroscopicity and CCN activity is more important with respect to smaller particles. However, it still remained a big challenge for directly obtaining surface tension of submicron particles. To our knowledge, there is no reported results that obtained surface tension of particles with diameter smaller than 100 nm. The size of collected particles for surface tension measurement using AFM are usually ranged in 0.5-1μm (Morris et al., 2015; Lee et al., 2017), which is limited by the diameter of nanoneedles. The diameter of High Aspect Ratio (HAR) nanoneedles should be smaller than 50 nm to detect surface tension of 100 nm particles. In our study, we used focused ion beam (FIB) induced deposition to produce needles with constant diameter (Ding et al., 2022). This method is difficult to make nanoneedles with diameter smaller than 100 nm. We are trying to improve our method in making AFM probe to measure the surface tension of smaller particles in the future.

2) Moreover, in this study, the $\kappa$ closure was conducted by measuring the pure and mixed inorganics and organics. However, the authors measured within the size range of ~50-260 nm as given in Table 2, which is different with the size range when they used for measuring surface tension. So, can the bias between $\kappa_{\text{CCN}}$ and $\kappa_{\text{Chem}}$ based on the larger size explain the effect of surface tension reduction? The surface tension should also dependent on particle size.

**Response: Thank you for the comment. We agree that the surface tension is dependent on particles size. Indeed, we admit that the discrepancy of particle diameter between surface tension and CCN activity experiments. However, we think this is not the main reason to explain the bias between $\kappa_{\text{CCN}}$ and $\kappa_{\text{Chem}}$.**

**1) As demonstrated by Cheng et al. (2015), the size dependence of interfacial energy will not play a significant role unless the solution droplets are smaller than 5 to 6 nm. Thus, the discrepancy of particle diameter between CCN experiment and AFM measurement may not affect our results.**

**2) Small droplet presents a higher surface-to-volume ratio than that of large particle (Ruehl and Wilson, 2014). Sorjamaa et al. (2004) and Miles et al. (2019) demonstrated that larger fraction of surface-active molecules in the small droplet may partition to the surface and surface tension reduction in small droplet possibly be stronger than large droplet.**

**Revision/addition:**

**1) Part of section 2.1.3, L119: "…However, it should be noted that the potential uncertainty introduced due to the different particle diameter in CCN activity (ranged in 50~260 nm) and AFM experiments (0.4-1μm) is not taken into account, because**

**the size dependence of surface tension is not significant unless the solution droplets are smaller down to 6 nm (Cheng et al., 2015).**"

3) In addition, it is not very clear that how did the author get the $\kappa$ values of each individual pure component, some more details are suggested to include in the Method.

**Response: Thank you for the suggestion. We have added the corresponding descriptions in Section 2.2.**

**Revision/addition:**

**Part of section 2.2, L124:** "Based on $\kappa$-Köhler theory, hygroscopicity parameter $\kappa_{CCN}$ **for individual pure component and mixed aerosol** can be calculated by:

$$\kappa_{CCN} = \frac{4A^3}{27D_d^3 ln^2(1+s_c)}, A = \frac{4M_w \sigma_w}{RT\rho_w} \tag{1}$$

where $\sigma_w$, $M_w$ and $\rho_w$ are surface tension, molecular weight and density of water, respectively. $R$ is universal gas constant and $T$ is temperature (298.15K). $s_c$ is critical supersaturation ratio. $D_d$ is dry diameter. In addition, hygroscopicity $\kappa$ of multicomponent chemical system can also be calculated assuming a Zdanovskii, Stokes, and Robinson (ZSR) simple mixing rule. $\kappa$ based on the chemical composition ($\kappa_{Chem}$) of mixed aerosol was calculated by:

$$\kappa_{Chem} = OVF \cdot \kappa_{org,CCN} + (1 - OVF) \cdot \kappa_{inorg,CCN} \quad, \tag{2}$$

**where $\kappa_{org,\ CCN}$ and $\kappa_{inorg,\ CCN}$ are obtained $\kappa_{CCN}$ values of single organic acid and inorganic salt.**"

4) Also, it is mentioned in the paper that the surface tension lowering is jointly determined by solubility, deliquescence RH and surface activity, and the influence of

solubility is greater. This conclusion is based on the data results that the surface tension of the two dicarboxylic acids (AA and OA) with low solubility decreases significantly at most. At the same time, these two dicarboxylic acids also have the higher deliquescence point and the longer carbon chain. However, the influence of deliquescence point on surface tension has not been discussed in this paper. As mentioned earlier, the particles are already liquid when measuring, that is, fully deliquescence. Is there a causal relationship between the deliquescence point and the surface tension lowering? In addition, does the carbon chain length mean that the surface activity must be strong? The authors may refer some previous studies to further clarify this.

**Response: Sorry for the misunderstanding.**

**1) We have realized that it is inappropriate to indicate the relation between surface activity and deliquescence RH based on our results. Therefore, we have removed the corresponding descriptions.**

**2) We have added the corresponding references to demonstrate that the surface activity of dicarboxylic acid become strong with increased carbon number.**

**Revision/addition:**

**Part of Section 3.4.2, L267:** "…Besides, notable surface tension reductions of particles containing OA or AA indicated that organic solubility plays an important role in surface tension reduction as AA and OA have the lowest solubilities among studied dicarboxylic acids. **OA and AA own higher carbon numbers than most of the rest studied organics. Since Aumann et al. (2010) found that the surface activity of dicarboxylic acids increases with carbon number from 2 to 9 based on surface tension measurement of their water solutions, indicating that dicarboxylic acids (e.g. OA and AA) with higher carbon number own stronger surface activity. Therefore, strong**

**surface activity of dicarboxylic acid is another factor attributing to surface tension reduction of inorganic salts/dicarboxylic acids.**"

**Conclusion:**

**1) L286:** "…**Slightly dissolved dicarboxylic acids with lower solubilities and strong surface activity are more likely to cause notable surface tension depression for inorganic salt/dicarboxylic acid mix particles.**"

**2) L289:** "…**especially for slightly soluble organics with lower solubilities and strong surface activity.**"

**References**

Aumann, E., Hildemann, L. M., and Tabazadeh, A.: Measuring and modeling the composition and temperature-dependence of surface tension for organic solutions, Atmos. Environ., 44, 329-337, https://doi.org/10.1016/j.atmosenv.2009.10.033, 2010.

Cheng, Y. F., Su, H., Koop, T., Mikhailov, E., and Poschl, U.: Size dependence of phase transitions in aerosol nanoparticles, Nat. Commun., 6, https://doi.org/10.1038/ncomms6923, 2015.

Ding, X., Kuang, B., Xiong, C., Mao, R., Xu, Y., Wang, Z., and Hu, H.: A Super High Aspect Ratio Atomic Force Microscopy Probe for Accurate Topography and Surface Tension Measurement, Sens. Actuators, A, 113891, https://doi.org/10.1016/j.sna.2022.113891, 2022.

Lee, H. D., Estillore, A. D., Morris, H. S., Ray, K. K., Alejandro, A., Grassian, V. H., and Tivanski, A. V.: Direct surface tension measurements of individual sub-micrometer particles using atomic force microscopy, J. Phys. Chem. A, 121, 8296-8305, https://doi.org/10.1021/acs.jpca.7b04041, 2017.

Miles, R. E. H., Glerum, M. W. J., Boyer, H. C., Walker, J. S., Dutcher, C. S., and Bzdek, B. R.: Surface Tensions of Picoliter Droplets with Sub-Millisecond Surface Age, J. Phys. Chem. A, 123, 3021-3029, https://doi.org/10.1021/acs.jpca.9b00903, 2019.

Morris, H. S., Grassian, V. H., and Tivanski, A. V.: Humidity-dependent surface tension measurements of individual inorganic and organic submicrometre liquid particles, Chem. Sci., 6, 3242-3247, https://doi.org/10.1039/c4sc03716b, 2015.

Ruehl, C. R. and Wilson, K. R.: Surface organic monolayers control the hygroscopic growth of submicrometer particles at high relative humidity, J. Phys. Chem. A, 118, 3952-3966, https://doi.org/10.1021/jp502844g, 2014.

Sorjamaa, R., Svenningsson, B., Raatikainen, T., Henning, S., Bilde, M., and Laaksonen, A.: The role of surfactants in Kohler theory reconsidered, Atmos. Chem. Phys., 4, 2107-2117, https://doi.org/10.5194/acp-4-2107-2004, 2004.

---

## Author Comment (AC2)

**Reply to comments on "Insights into the role of dicarboxylic acid on CCN activity: implications for surface tension and phase state effects" by Xiong et al.**

**Reply to Anonymous Referee #2**

1) The manuscript by Xiong et al. discussed the impact of surface tension reduction on CCN activity of dicarboxylic acid-inorganic salt mixtures. CCN activity was quantified using the CCN counter, and surface tension was measured using the AFM. The data suggested that observed kappa values for adipic acid (AA) and octanedioic acid (OA) cannot be well explained by chemical composition when surface tension of water is assumed. The AFM data demonstrated that the values of surface tension for these particles were significantly lower than that of water. The result makes sense, and the output of the study will be useful for future studies on CCN activity. I have some comments on the manuscript that needs to be considered for making it to be acceptable to the journal. I also suggest the authors to ask a native speaker of English for checking grammatical issues on the manuscript.

Response: We truly appreciate the constructive comments and suggestions raised by the reviewer. Those comments are valuable and very helpful for improving our paper, as well as the important guiding significance to our studies. Below we provide a point-by-point response to individual comment. The responses are shown in brown and bold fonts, and the added/rewritten parts are presented in blue and

**bold fonts. Also, we have asked native speaker of English for checking grammatical issues and the revised manuscript has been improved.**

**Major comments:**

2) The manuscript qualitatively connected reduction in surface tension and kappa. However, these two parameters are not quantitatively connected in the current manuscript. For instance, it would be possible to develop a multicomponent Kohler model considering water-solubility of pure organic compounds, and investigate sensitivity of the measured kappa values on the assumed value of surface tension. If the measured values of surface tension can explain the experimentally constrained value of kappa, this study could be more quantitative. The reviewer would imagine that the quantitative study could have been easier if information about particle water contents were to be available for the AFM data. Would the authors provide comment on it?

**Response: Thank you for the suggestion. Actually, we could not obtain water content from AFM data. But it is a very good idea to constrain the water solubility and surface tension in Köhler model. Here, we used the solubility-involved Köhler model which was introduced by Petters and Kreidenweis (2008), to investigate sensitivity of the measured $\kappa_{CCN}$ values on the assumed value of surface tension for inorganic salts/OA systems (results for inorganic salts/AA were not shown here because there were only two measured surface tension results).**

**As shown in Figure 1a, $\kappa_{CCN}$ of NaCl/OA with 60%, 75% and 89% OVF derived from solubility-involved Köhler model (circles) with water surface tension were 0.515, 0.324 and 0.145. These values underpredict our $\kappa_{CCN}$ base on CCN measurement (0.688, 0.485 and 0.296, triangles). However, if modeled $\kappa_{CCN}$ values fit the measured**

values, the corresponding surface tensions should reduce to 65.4 mN m$^{-1}$ (60% OVF), 62.7 mN m$^{-1}$ (75% OVF), 56.7 mN m$^{-1}$ (89% OVF). Similar results were also found for AS/OA systems (Fig.1b).

In Fig. 1c, fitted surface tension showed good linear relation with measured surface tensions (slope and R$^2$ is 1.01 and 0.71, respectively.). This could provide a quantitate way to predict $\kappa_{CCN}$ values of inorganic salts/OA by solubility-involved Köhler model, by using their measured surface tensions results.

[Figure]

**Fig. 1** $\kappa_{CCN}$ vs. assumed surface tension for (a) NaCl/OA and (b) AS/OA systems according to solubility-involved Köhler model presented by Petters and Kreidenweis (2008). The triangles and circles in (a) and (b) represent the measured $\kappa_{CCN}$ and predict $\kappa_{CCN}$ by solubility-involved Köhler model. Closure between

fitted surface tensions and measured surface tensions (c). $\sigma_w$ represents water surface tension (72 mN m$^{-1}$).

**Minor comments:**

1) **Title:** Researchers in the area already know that dicarboxylic acids are important contributors to CCN. It is better to stress the novelty of the study in the title better.

**Response: Thank you for the suggestion. We have revised our title.**

**Revision/addition:**

**Title: "Reconsideration of surface tension and phase state effects on CCN activity based on the AFM measurement."**

2) **Line 36**: Add references to support the statement.

**Response: Thanks, we have added references.**

**Revision/addition:**

**Line 37: The role of particle chemistry in the activation process, however, is still debatable due to the complexity of chemical constitution (Bhattu and Tripathi, 2015; Noziere, 2016).**

3) **Section 2.1.2**: qualities of the compressed air and water for the atomizer are very important for CCN activity studies of acidic chemical species. Ammonia ubiquitously exists in an indoor environment. Based on the reviewer's experience, it has never been easy to generate ammonia-free dicarboxylic acid particles. I suggest adding further details about particle generation in the revised manuscript.

**Response: Thank you for the advice. We have added the details of water information in section 2.1.2. For the indoor ammonia, we agree the possible influence of indoor ammonia during the generation of acid particles. But the consistence of $\kappa_{CCN}$ between our results and previous studies (e.g. malonic acid, succinic acid and adipic acid) implied that this influence might be ignored.**

**Revision/addition:**

**Line 82: In brief, particles containing single and mixed chemicals were generated by clean and particle-free compressed air with water solutions (~ 1‰) by a constant output atomizer (TSI 3079A). The solutions were prepared by using ultrapure water (Millipore, resistivity $\leq$ 18.2M$\Omega$).**

4) **Line 93:** Is there any reason why the authors selected the hydrophobic silicon wafer as a substrate? What would be the advantages/disadvantages of the substrate when compared with other types of substrate?

**Response: The hydrophobic silicon wafer was frequently used to measure single particle's surface tension using AFM (Morris et al., 2015; Lee et al., 2017; Lee et al., 2020). The different types of substrate results in different affinity between water and substrate. Hydrophobic silicon wafer used in our study has been proved that almost all of the solute can be collected into the solute aggregate on the surface after water evaporation (Ding et al., 2020). Therefore, our hydrophobic silicon wafer removes the possible loss of solute when RH varies (especially RH decreases), which could not be ensured by other types of substrate such as commonly used mica sheet and normal silicon wafer with no hydrophobic coatings on surface.**

**Revision/addition:**

**Line 97:** "**particle sizing sampler (Models BGI20800 Series, BGI Incorporation) onto hydrophobically silicon wafers.** The hydrophobically silicon wafers are with polydimethylsiloxane brush surface, so solute can be collected into the solute aggregate on the surface after water evaporation when RH varies (especially RH decreases) (Ding et al., 2020)."

5) **Line 99:** I checked the datasheet of SHT 85. The accuracy of the sensor is +-1.5%. The authors mentioned in the manuscript that the AFM measurement was conducted at 99.5% of RH. The uncertainty of 1.5% for the high RH region influences significantly influences thermodynamic properties. The potential influence of uncertainties in RH measurements on the AFM data would need to be discussed in detail.

**Response: Thank you for the comment. We agree that under such high RH level, ± 1.5% accuracy may bring uncertainties in droplet water content and surface tension results. Therefore, we successively obtained more than 10 force plots of at least 5 individual droplets in within 5~10 minutes to decreased the uncertainties as much as possible. In our study, most of the results (97%) showed standard deviation within 10%, showing a relative low effect on surface tension results. Especially for inorganic salts mixed with MA, PhMA, SA, PhSA and PA, their standard deviations are even lower than 6%, indicating a negligible influence of RH sensor accuracy in this study.**

**Revision/addition:**

**Line 104:** "was achieved and maintained by humidified flow. RH in cell was measured by a RH sensor (SHT 85, ± 1.5% uncertainty, Sensirion Inc.)."

**Line 116:** "More than 10 force plots were collected on at least 5 individual droplets in order to decreased the uncertainties (e.g. sensor accuracy)".

6) **Line 103:** Ideally, this manuscript should be published after the publication of the procedure for making the nanotip. At least, the name of the potential first author needs to be stated so that the readers will be able to search for the corresponding paper if necessary.

**Response: Thank you for reminding. The manuscript about making nanoneedle has been published and the reference has been added in the revised manuscript.**

**Revision/addition:**

**Line 107: "… The procedures of making nanotips were detailly described in Ding et al. (2022) and a brief description was given here.**

7) **Line 122:** OVF needs to be defined.

**Response: The definition has of OVF has been added.**

**Revision/addition:**

**Line 132: OVF indicates the organic volume fraction of mixed particles.**

8) **Line 133:** Although the authors mention that the data are consistent with previous results, they are actually slightly different, as discussed in the following sentences in the same paragraph. The expression should be updated for representing the comparison more accurately.

**Response: We have added more descriptions about the comparison as suggested.**

**Revision/addition:**

**Line 141:** "$\kappa_{CCN}$ values for single component aerosols were summarized in Table 2. $\kappa_{CCN}$ of NaCl, AS, MA, SA and AA were 1.325 ± 0.038, 0.562 ± 0.059, 0.240 ± 0.036, 0.204 ± 0.023 and 0.008 ± 0.001, respectively, being **overall** consistent with previous results (Petters and Kreidenweis, 2007; Kuwata et al., 2013). **$\kappa_{CCN}$ of NaCl and MA were slightly higher while AS was slightly lower than those reported in Petters and Kreidenweis (2007). This may be ascribed to the solute purity (Hings et al., 2008). Based on the same reason, $\kappa_{CCN}$ of PA (0.112 ± 0.010) and OA (0.003 ± 0.0002) were 20% lower and twice higher than those reported by Kuwata et al. (2013), respectively.**"

9) **Line 144:** It is not clear to me how the criteria for highly- and slightly- soluble compounds were developed, and why it is important for the present study. Further information is needed.

**Response: Sorry for the misunderstanding. The criteria for highly and slightly soluble compounds was set to 100 g/ L according to Kuwata et al. (2013) and Luo et al. (2020). The $\kappa_{CCN}$ values for highly soluble components (MA, PhMA and PhSA) displayed no monotone trend with solubilities, while $\kappa_{CCN}$ values of sparely soluble components (AA, PA, SA and OA) showed an increased trend with solubility.**

**Revision/addition:**

**Line 148:** "Solubility and molar volume of dicarboxylic acids were essential factors influencing their hygroscopicity (Kumar et al., 2003; Han et al., 2022). **Therefore, solubility criteria of 100 g/L was used in our study to distinguish the effect of**

solubility of highly soluble (with water solubility over 100 g L$^{-1}$) and slightly soluble organics (with water solubility below 100 g L$^{-1}$) on their hygroscopicity, according to Kuwata et al. (2013) and Luo et al. (2020)."

10) **Line 172-174:** I could not understand what this long sentence means well. It would be great if the authors could update the description.

Response: We are sorry for misunderstanding, we have revised the description.

Revision/addition:

Line 182: "Thus, addition of inorganic salts facilitates deliquescence of OA and AA under lower RH, further promotes their phase state transition from solid to liquid (or semisolid), and their surface tension would be reduced. Based on surface tension results of water solutions, Aumann et al. (2010) reported that surface activities of dicarboxylic acids were increased with their carbon number. Therefore, surface tensions of inorganic salts/AA and inorganic salts/OA may decrease more than the rest acids containing particles, resulting in their relatively higher $\kappa_{CCN}$. This indication was further confirmed by AFM surface tension measurement, as discussed in Section 3.4."

11) **Table 1:** What does 'guaranteed reagent' mean? Would you provide the detailed information about what is specifically guaranteed?

Response: Detailed information has been provided.

Revision/addition:

[revised manuscript text omitted]

---

## Author Response (AR2)

**Reply to comments on "Reconsideration of surface tension and phase state effects on CCN activity based on the AFM measurement" by Xiong et al.**

**Reply to Anonymous Referee #2**

**Minor comments:**

1) Authors revised the manuscript well by considering the most of the comments by the reviewers. The authors provided an excellent response to the major comment by reviewer #2. However, the content is not reflected to the revised manuscript. It should make sense to add the contents to the revised manuscript unless the authors have a significant concern about it. I believe that the revised manuscript will meet the criteria of the journal after adding the content to the manuscript.

**Response: We really appreciate the constructive comments and suggestions raised by the reviewer. As suggested, we have added the response of the major comment to our revised manuscript.**

**Addition:**

**L273: In order to quantitively connect surface tension and measured $\kappa_{CCN}$, we used the solubility-involved Köhler model which was introduced by Petters and Kreidenweis (2008), to investigate sensitivity of the measured $\kappa_{CCN}$ values on the assumed value of surface tension for inorganic salts/OA systems. As shown in Fig. 7a,**

$\kappa_{CCN}$ of NaCl/OA with 60%, 75% and 89% OVF derived from solubility-involved Köhler model with water surface tension were 0.515, 0.324 and 0.145 (circles). These values underpredict measured $\kappa_{CCN}$ (0.688, 0.485 and 0.296, triangles). However, if modeled $\kappa_{CCN}$ values fit the measured values, the corresponding surface tensions should reduce to 65.4 mN m$^{-1}$ (60% OVF), 62.7 mN m$^{-1}$ (75% OVF), 56.7 mN m$^{-1}$ (89% OVF). Similar results were also found for AS/OA systems (Fig.7b). In Fig. 7c, fitted surface tension showed good linear relation with measured surface tensions (slope and R$^2$ were 1.09 and 0.71). Therefore, our results could provide a quantitative way to predict $\kappa_{CCN}$ values of inorganic salts/OA based on solubility-involved Köhler model, by using their measured surface tensions results under high RH. This quantitative method should be tested for more chemical systems in the future.

[Figure]

**Fig. 7** $\kappa_{CCN}$ vs. assumed surface tension for (a) NaCl/OA and (b) AS/OA systems according to solubility-involved Köhler model presented by Petters and Kreidenweis (2008). The triangles and circles in (a) and (b) represent the measured $\kappa_{CCN}$ and predict $\kappa_{CCN}$ by solubility-involved Köhler model. Closure between fitted surface tensions and measured surface tensions (c). $\sigma_w$ represents water surface tension (72 mN m$^{-1}$).

**Reference**

Petters, M. D., and Kreidenweis, S. M.: A single parameter representation of hygroscopic growth and cloud condensation nucleus activity - Part 2: Including solubility, Atmos. Chem. Phys., 8, 6273-6279, 10.5194/acp-8-6273-2008, 2008.